# Northern Hemisphere blocking simulation in current climate models: evaluating progress from CMIP5 to CMIP6 and sensitivity to resolution

Reinhard Schiemann[1], Panos Athanasiadis[2], David Barriopedro[3], Francisco Doblas-Reyes[4,5], Katja Lohmann[6], Malcolm J. Roberts[7], Dmitry Sein[8], Christopher D. Roberts[9], Laurent Terray[10], and Pier Luigi Vidale[1]

[1]National Centre for Atmospheric Science, Department of Meteorology, University of Reading, Reading, United Kingdom
[2]Euro-Mediterranean Centre for Climate Change (CMCC), Bologna, Italy
[3]Instituto de Geociencias (IGEO), CSIC-UCM, Madrid, Spain
[4]Catalan Institution for Research and Advanced Studies (ICREA), Barcelona, Spain
[5]Earth Sciences Departement, Barcelona Supercomputing Center (BSC), Barcelona, Spain
[6]Max Planck Institute for Meteorology, Hamburg, Germany
[7]Met Office Hadley Centre, Exeter, United Kingdom
[8]Alfred Wegener Institute, Bremerhaven, Germany
[9]European Centre for Medium Range Weather Forecasting (ECMWF), Reading, United Kingdom
[10]Climat, Environnement, Couplages, Incertitudes, CECI, Université de Toulouse, CNRS, Cerfacs, Toulouse, France

**Correspondence:** Reinhard Schiemann (r.k.schiemann@reading.ac.uk)

**Abstract.** Global Climate Models (GCMs) are known to suffer from biases in the simulation of atmospheric blocking, and this study provides an assessment of how blocking is represented by the latest generation of GCMs. It is evaluated (i) how historical CMIP6 (Climate Model Intercomparison Project Phase 6) simulations perform compared to CMIP5 simulations, and (ii) how horizontal model resolution affects the simulation of blocking in the CMIP6-HighResMIP (PRIMAVERA) model ensemble, which is designed to address this type of question. Two blocking indices are used to evaluate the simulated mean blocking frequency and blocking persistence for the Euro-Atlantic and Pacific regions in winter and summer against the corresponding estimates from atmospheric reanalysis data. There is robust evidence that CMIP6 models simulate blocking frequency and persistence better than CMIP5 models in the Atlantic and Pacific and during winter and summer. This improvement is sizeable so that, for example, winter blocking frequency in the median CMIP5 model in a large Euro-Atlantic domain is underestimated by 33% using the absolute geopotential height (AGP) blocking index, whereas the same number is 18% for the median CMIP6 model. As for the sensitivity of simulated blocking to resolution, it is found that the resolution increase, from typically 100 km to 20 km grid spacing, in most of the PRIMAVERA models, which are not re-tuned at the higher resolutions, benefits the mean blocking frequency in the Atlantic in winter and summer, and in the Pacific in summer. Simulated blocking persistence, however, is not seen to improve with resolution. Our results are consistent with previous studies suggesting that resolution is one of a number of interacting factors necessary for an adequate simulation of blocking in GCMs. The improvements reported in this study hold promise for further reductions in blocking biases as model development continues.

# 1 Introduction

Atmospheric blocking refers to the occurrence of quasi-stationary high pressure systems in the middle and high latitudes. Blocking highs persist for several days to weeks and divert cyclones pole- or equatorward. Preferred regions of blocking occurrence are the eastern sides of the Atlantic and Pacific basins, and blocking events occur throughout the year. Blocking is associated with anomalous surface weather conditions such as cold spells in winter and heat waves in summer, and these surface impacts can be hazardous especially for long-lasting blocking events (e.g., Rex, 1950; Barriopedro et al., 2010; Cattiaux et al., 2010; Woollings, 2010; Barriopedro et al., 2011; Matsueda, 2011; Otto et al., 2012).

Global climate models (GCMs) tend to underestimate blocking, and these biases have been long standing and documented for different phases of the Atmospheric/Coupled Model Intercomparison Project (AMIP/CMIP) efforts (D'Andrea et al., 1998; Masato et al., 2013; Anstey et al., 2013; Davini and D'Andrea, 2016). Too coarse horizontal resolution of the atmospheric grid has been put forward as one of the factors limiting the accurate represenation of blocking in GCMs and several studies show how an increase in resolution benefits the simulation of blocking (Matsueda et al., 2009; Anstey et al., 2013; Davini et al., 2017; Schiemann et al., 2017). At the same time, the magnitude of the improvements seen at higher resolution varies considerably between different studies due to issues including the GCM(s) considered, their resolutions, and blocking indices used for evaluation as well as the ensemble size employed and simulation period covered to sample internal variability of blocking. For example, Matsueda et al. (2009) report a dramatic improvement in the simulated Euro-Atlantic blocking as resolution is increased from 180 km to 20 km in an atmospheric GCM (AGCM), whereas Anstey et al. (2013) and Schiemann et al. (2017) show more modest improvements with resolution across the CMIP5 simulations, and in a 4-model ensemble of AMIP simulations with grid spacings down to 20 km, respectively. Also, Davini et al. (2017) show that in their AGCM a very good representation of Euro-Atlantic blocking at grid spacings of 40 km or smaller is the result of compensating biases between too strong eddies at upper and too weak eddies at lower levels. In addition to horizontal resolution, a number of other factors have been shown to be important for blocking simulation including sea surface temperature and associated mean-state biases, vertical resolution, orographic boundary conditions, physcial parameterisations, and the numerical scheme employed in the model's dynamical core (Woollings et al. (2018) and references therein including Scaife et al. (2011), Anstey et al. (2013), Berckmans et al. (2013), Jung et al. (2012), Pithan et al. (2016), Williams et al. (2018)).

The aim of this paper is twofold: First, recently available model simulations following the CMIP6-HighResMIP protocol, delivered by the EU Horizon 2020 PRIMAVERA project, offer a new set of controlled experiments designed to address the sensitivity to model resolution in a coordinated multi-model ensemble (see Sect. 2.1). Second, we adopt a slightly longer-term view and compare blocking biases in the latest generation of high-resolution HighResMIP GCMs with those seen in historical CMIP6 and CMIP5 simulations, which serves to put the HighResMIP results in context and appears timely given both the very recent availability of CMIP6-HighResMIP simulations and the potential need for up-to-date model evaluations underpinning, for example, the Intergovernmental Panel on Climate Change (IPCC) Sixth Assessment Report. We conduct all evaluations for two different blocking indices so as to assess the robustness of our results with respect to this choice. In addition to evaluating

the mean blocking climatology, we also evaluate the representation of blocking persistence, which appears to be missing from the recent literature.

The structure of this paper is as follows: In Sect. 2, we introduce the multi-model ensembles that are evaluated in this study and the reference reanalysis blocking climatology. We also introduce the two blocking indices and describe how the persistence analysis is conducted for each of these indices. The three following sections report on the results of our evaluations, namely the spatial distribution of simulated blocking and biases in Sect. 3, the quantification of domain-mean blocking biases and how they depend on model resolution in Sect. 4, and the evaluation of blocking persistence in Sect. 5. The paper is concluded in Sect. 6.

## 2 Data and methods

### 2.1 Multi-model ensembles and experiments

We use the simulations delivered by the PRIMAVERA (PRocess-based climate sIMulation: AdVances in high resolution modelling and European climate Risk Assessment) project to assess the sensitivity of simulated blocking to horizontal model resolution. These simulations follow the CMIP6-HighResMIP protocol (Haarsma et al., 2016) and are designed to test how the simulation of a range of phenomena in the climate system depends on model resolution in the atmosphere and ocean. The high-resolution versions of the PRIMAVERA models have therefore been re-tuned as little as possible with respect to their low-resolution counterparts. This is in line with the HighResMIP philosophy that prioritises being able to attribute any changes in model performance to the direct effect of resolution change over building optimally tuned models with as small as possible biases. For the evaluation of how simulated blocking is sensitive to resolution presented here, this implies that our results should be considered to be conservative as mean-state circulation biases are known to statistically explain a large part of, yet not all, blocking biases seen in climate models (Scaife et al., 2010; Vial and Osborn, 2012; Schiemann et al., 2017).

The PRIMAVERA models and simulations used in this study are listed in Table 1. The simulation period is 1950–2014 and we evaluate historical simulations driven by observed greenhouse-gas and aerosol concentrations both in a coupled ocean-atmosphere-land-sea ice setup (HighResMIP *hist-1950* experiment) and in an AMIP-style setup driven by historically observed sea-surface temperature and sea-ice concentrations (HighResMIP *highresSST-present* experiment). Further details of the experimental setup and a baseline evaluation focusing on coupled aspects of the HadGEM3-GC3.1 model are provided by Roberts et al. (2019).

In addition to the PRIMAVERA simulations, we evaluate the representation of blocking in one historical simulation with each of 29 CMIP5 models for the period 1950–2005 (Taylor et al., 2012) covering the period 1950–2005, and for one historical simulation with each of 13 CMIP6 models for the period 1950–2014 (Eyring et al., 2016). The CMIP5 and CMIP6 model resolutions correspond to grid spacings between about 100 and 400 km, and 100 and 300 km, respectively.

**Table 1.** PRIMAVERA models and simulations. Columns detail the model name, the atmosphere grid spacing at $50°$N, nominal ocean grid spacing, a sub-ensemble indicator ('LF' — low-resolution forced (AMIP), 'LC' — low-resolution coupled, 'HF' — high-resolution forced, and 'HC' — high-resolution coupled), the number of ensemble members used in this study, and model documentation references.

| No. | Model | Atm. grid (km) | Ocean grid (km) | Sub-ensemble | Members | References |
|---|---|---|---|---|---|---|
| 1 | AWI-CM-1-1-LR | 129 | 50 | LC | 1 | |
| 2 | AWI-CM-1-1-HR | 67 | 25 | HC | 1 | Sein et al. (2017) |
| 3 | CMCC-CM2-HR4 | 64 | - | LF | 1 | |
| 4 | CMCC-CM2-VHR4 | 18 | - | HF | 1 | |
| 5 | CMCC-CM2-HR4 | 64 | 25 | LC | 1 | Cherchi et al. (2019) |
| 6 | CMCC-CM2-VHR4 | 18 | 25 | HC | 1 | |
| 7 | CNRM-CM6-1 | 142 | - | LF | 1 | |
| 8 | CNRM-CM6-1-HR | 50 | - | HF | 1 | |
| 9 | CNRM-CM6-1 | 142 | 100 | LC | 1 | Voldoire et al. (2019) |
| 10 | CNRM-CM6-1-HR | 50 | 25 | HC | 1 | |
| 11 | EC-Earth3P | 71 | - | LF | 1 | |
| 12 | EC-Earth3P-HR | 36 | - | HF | 2 | |
| 13 | EC-Earth3P | 71 | 100 | LC | 3 | Haarsma et al. (2020) |
| 14 | EC-Earth3P-HR | 36 | 25 | HC | 3 | |
| 15 | ECMWF-IFS-LR | 50 | - | LF | 8 | |
| 16 | ECMWF-IFS-HR | 25 | - | HF | 6 | |
| 17 | ECMWF-IFS-LR | 50 | 100 | LC | 8 | Roberts et al. (2018) |
| 18 | ECMWF-IFS-MR | 50 | 25 | LC | 3 | |
| 19 | ECMWF-IFS-HR | 25 | 25 | HC | 6 | |
| 20 | HadGEM3-GC31-LM | 135 | - | LF | 5 | |
| 21 | HadGEM3-GC31-MM | 60 | - | HF | 3 | |
| 22 | HadGEM3-GC31-HM | 25 | - | HF | 3 | Roberts et al. (2019), |
| 23 | HadGEM3-GC31-LL | 135 | 100 | LC | 8 | Williams et al. (2018), |
| 24 | HadGEM3-GC31-MM | 60 | 25 | HC | 1 | Kuhlbrodt et al. (2018), |
| 25 | HadGEM3-GC31-HM | 25 | 25 | HC | 3 | Menary et al. (2018) |
| 26 | HadGEM3-GC31-HH | 25 | 8 | HC | 1 | |
| 27 | MPI-ESM1-2-HR | 67 | - | LF | 1 | |
| 28 | MPI-ESM1-2-XR | 34 | - | HF | 1 | |
| 29 | MPI-ESM1-2-HR | 67 | 40 | LC | 1 | Gutjahr et al. (2019) |
| 30 | MPI-ESM1-2-XR | 34 | 40 | HC | 1 | |

## 2.2 Observed blocking

The reference data for evaluating model-simulated blocking is based on both the ERA-40 (Uppala et al., 2005) and ERA-Interim reanalyses (Dee et al., 2011). Following Schiemann et al. (2017), we concatenate data from these two reanalyses to obtain a 50-year reference climatology covering the period 1962–2011 so as to reduce the impact of blocking internal variability

on our results. Schiemann et al. (2017) also show that these two reanalyses, as well as the MERRA reanalysis (Rienecker et al., 2011), agree very well with each other on the mean and interannual variability of blocking over different domains implying that for the purposes of this paper reanalysis uncertainty can be considered to be small compared to internal variability.

## 2.3 Blocking indices

A considerable number of blocking indices have been employed by different authors, and these indices emphasise different aspects of the blocking phenomenon and use different meteorological variables (see, e.g., Barriopedro et al., 2010, for an overview) so that it is advantageous to use more than one blocking index to assess the robustness of model evaluation results to this choice (e.g., Woollings et al., 2018). One fundamental distinction is between blocking indices based on the exceedance of an absolute (fixed) threshold of a meteorological variable and indices based on the detection of anomalies (departures) of a meteorological variable from a climatological mean. Here, we use one index from each of these two groups, namely the so-called absolute geopotential height (AGP) index described in Sect. 2.3.1 and the anomaly index (ANOM) described in Sect. 2.3.2. We calculate both of these indices from daily-mean 500hPa geopotential height data for the simulations and reanalysis data introduced in Sections 2.1 and 2.2.

### 2.3.1 Absolute Geopotential Height index

The AGP index is a generalisation of the one-dimensional index by Tibaldi and Molteni (1990) to two dimensions (Scherrer et al., 2006). According to the AGP index, three conditions need to be fulfilled for a point at latitude $\phi_0$ to be identified as blocked. The first condition is a reversal of the climatological equator-to-pole gradient of the 500hPa geopotential height $Z$ to the south of $\phi_0$:

$$\frac{Z(\phi_0) - Z(\phi_S)}{\phi_0 - \phi_S} > 0 \,, \tag{1}$$

where $\phi_S$ is 15° south of $\phi_0$. The second condition requires westerlies to the north of $\phi_0$:

$$\frac{Z(\phi_N) - Z(\phi_0)}{\phi_N - \phi_0} < -10 \, \mathrm{m} \, (°\mathrm{latitude})^{-1} \,, \tag{2}$$

where $\phi_N$ is 15° north of $\phi_0$. The third condition is that the point is only considered blocked if the first two conditions are met for five consecutive days or more. All model and reanalysis fields are regridded to a common $1.875° \times 1.25°$ grid before the blocking identification is applied, and we calculate the blocking index for all grid boxes between 35°N and 75°N. This index has been used in previous evaluations of blocking in multi-model ensembles (Anstey et al., 2013; Schiemann et al., 2017).

### 2.3.2 Anomaly index

The ANOM index (following Woollings et al. (2018) and similar to Sausen et al. (1995) and Schwierz et al. (2004) but using 500hPa geopotential height Z500) is based on tracking geopotential height anomalies. The following steps are carried out in its calculation:

1. Daily Z500 data are regridded to a common 2.5° grid and a 31-day running mean is calculated through a baseline period (1981–2010) and a daily Z500 climatology is obtained by taking the mean over the baseline period for each day.

2. A daily anomaly is calculated, separately for each month, by taking the difference between the original Z500 data and the climatology from step 1 for the corresponding day. A monthly, spatially invariable, anomaly threshold is then obtained by calculating the 90th percentile of these differences throughout 50–80°N. The monthly anomaly threshold is smoothed further with a 3-month rolling mean.

3. For each day, potential blocking events are identified as contiguous areas of at least $2 \times 10^6$ km$^2$ where the Z500 anomaly (as in step 2) exceeds the monthly anomaly threshold (also as in step 2).

4. The candidate events from step 3 are further screened by requiring a spatial overlap of at least 50% between consecutive days (quasi-stationarity) for at least 5 days (minimum persistence).

## 2.4  Spatial aggregation and persistence analysis

Spatially averaged metrics of blocking performance are calculated for domains centered over the North Atlantic (ATL) and North Pacific (PAC), respectively. The definition of these domains reflects the spatial patterns of AGP and ANOM climatologies and biases (see Sect. 3) and we choose about the same ATL domain for both indices (-90–90°E, 50–75°N for AGP and -90–90°E, 50–90°N for ANOM) but somewhat different domains for PAC (90–270°E, 50–75°N for AGP and 120–240°E, 40–90°N for ANOM).

For the analysis of blocking persistence (Sect. 5), both blocking indices are used as described in Sections 2.3.1 and 2.3.2 but the persistence criterion of 5 days is relaxed so that blocking events of any persistence, including so-called instantaneous blocking events on a single day, which are not strictly considered to be blocks, are included in the analysis.

For the locally defined AGP index, defining the persistence of blocking at the grid-box scale is not meaningful and spatial aggregation is necessary before the persistence analysis is carried out. We aggregate here over 12 sectors of 30° longitude in 50–75°N and a sector is said to be blocked on a given day if at least 10% of the sector area is blocked according to the AGP index. Persistence analysis results shown for the ATL and PAC domains are then average results for the corresponding sectors. For the ANOM index based on tracking spatially extended geopotential height anomalies no such spatial aggregation is necessary and the persistence analysis is carried out using the persistence of these anomalies directly.

For both indices, we determine the empirical survival function ESF($t$), i.e. the probability of a blocking event to persist for at least $t$ days. Quantiles of ESF($t$) are estimated using the nonparametric Kaplan-Meier estimator for the AGP index, whereas for the ANOM index a parametric exponential fit was found to work well. See, e.g., Tableman et al. (2003), for details, noting that our application is much simpler than a typical survival analysis as there are no censored observations.

## 3 Geographical distribution of blocking occurrence

Maps of time mean winter (DJF) AGP blocking frequency are shown in Fig. 1. For an overview of the effect of resolution and coupling, the blocking climatology is shown separately for four PRIMAVERA sub-ensembles in Fig. 1a–d. In the sub-ensembles, simulations have been grouped into (i) low-resolution forced ('LF'), (ii) low-resolution coupled ('LC'), (iii) high-resolution forced ('HF'), and (iv) high-resolution coupled ('HC') simulations as shown in Table 1. Winter blocking climatologies for the CMIP5 and CMIP6 ensembles are shown in Fig. 1e,f.

All of the model ensembles considered show widespread underestimation of the climatological blocking frequency in the Pacific and especially in the Euro-Atlantic region. Theses ensemble-mean biases vary strongly in space, with an underestimation of several 10% being typical (see Sect. 4 for further quantification). These biases are also pervasive across the different ensembles as indicated by the stippling showing agreement among models on the sign of the blocking frequency biases. Closer inspection also shows differences between model ensembles. CMIP5 biases are spatially similar to those of the CMIP6 models, yet the multi-model mean CMIP6 bias is smaller than the CMIP5 bias throughout the Euro-Atlantic region, whereas this difference is smaller for the Pacific (Fig. 1e,f). Multimodel-mean biases for the high-resolution PRIMAVERA models are smaller than for the low-resolution PRIMAVERA models (Fig. 1a,b vs. Fig. 1c,d). This improvement with resolution is seen for both AMIP and coupled simulations over the Euro-Atlantic region, and also for the Pacific in the coupled simulations.

In summer, blocking is observed throughout a wide high-latitude region ranging from Greenland across northern Eurasia to Alaska (Fig. 2) so that the distinction between Atlantic and Pacific blocking is not as clear as in winter. As in winter, all model ensembles underestimate the blocking frequency everywhere and this bias is pervasive across models. There are small improvements in this bias in CMIP6 over CMIP5 in the Baltic region and over Siberia, with little change elsewhere (Fig. 2e,f). When comparing the different PRIMAVERA sub-ensembles, blocking is again seen to improve at the higher resolution, and this improvement is seen more clearly and includes the Pacific region in the coupled simulations (Fig. 2a–d).

Repeating the same analyses with the ANOM blocking index (Figures S1 and S2 in the supplement) shows results that largely agree with those based on the AGP index. All model ensembles are found to underestimate the occurrence of blocking both in the Euro-Atlantic and Pacific regions, and both in winter and in summer. There are also small improvements from CMIP5 to CMIP6 over the Atlantic in winter and summer, whereas the difference between the CMIP5 and CMIP6 biases is small for the Pacific. Small improvements with resolution are also seen in the PRIMAVERA ensemble, yet these appear smaller, as a fraction of total blocking frequency, than for the AGP index and these improvements are not seen for the Pacific.

## 4 Sensitivity to model resolution

We proceed in this section with a quantitative evaluation of how different metrics of blocking performance depend on atmospheric model resolution. Given the comparatively large biases seen in the Euro-Atlantic sector, we focus on the ATL domain and use the AGP index in the main manuscript (Figures 3 and 4) but also include results for the PAC domain and the ANOM index in the supplement (Figures S3–S8).

## 4.1 Atlantic

The evaluation for the ATL domain in winter using the AGP index is shown in Fig. 3. The six panels show different metrics of blocking performance (domain-mean blocking frequency — top row, spatial correlation with the reanalysis climatology across the domain — middle row, root-mean-square error with respect to the reanalysis climatology — bottom row), for PRIMAVERA coupled (left-hand panels) and AMIP simulations (right-hand panels). Each panel is divided into two parts; on the left the blocking metric is plotted vs. resolution for each of the PRIMAVERA models, using the ensemble-mean metric if more than one simulation is available for a given model/resolution (Table 1). On the right, the distribution of the same metric is shown in terms of a boxplot for each of the CMIP5 and CMIP6 ensembles, and the reanalysis estimate is shown by the '*' symbol labelled 'ERA/IV' on the x-axis. All estimates of metrics of blocking performance shown are subject to internal (sampling) variability and a model-based estimate of this internal variability is shown by the boxplot labelled 'ERA/IV'. These boxplot statistics are obtained by forming 15 pairs of simulations from the 6-member ensembles available for the ECMWF-IFS-HR model (No. 16 and 19 in Table 1) and quantifying the agreement on the blocking metric in these pairs.

Atlantic winter blocking is seen to be systematically underestimated by nearly all of the CMIP5, CMIP6, and PRIMAVERA simulations analysed (Fig. 3). For the ATL domain and AGP index chosen, this bias ranges from less than half of the observed blocking frequency for models simulating very little blocking to an underestimation of around 10-20% for models simulating more frequent blocking. There is a systematic shift from CMIP5 to CMIP6 models showing a better agreement of the CMIP6 models with reanalysis data for all three metrics considered. With the exception of one or two models, there is also a general tendency of the coupled PRIMAVERA simulations for an improved simulation, i.e. more frequent blocking, a higher spatial correlation, and a smaller root-mean-squared error, at the higher resolutions. This systematic improvement with resolution is not seen in the AMIP simulations for which the sensitivity of blocking performance to resolution as well as the variation in blocking performance between different models is smaller than for the coupled simulations. The AMIP simulations show similar or slightly better blocking performance than the corresponding coupled simulations for most models with the notable exception of the Hadley Centre model (HadGEM3-GC3.1). Repeating the same analysis with the ANOM index corroborates the pervasive underestimation of ATL winter blocking, by typically 40-10% with this index (Fig. S3). The improvement from CMIP5 to CMIP6 is also seen with the ANOM index. The improvement with resolution is not as clear as for the AGP index; noteworthy, though not statistically significant, is the fact that the RMSE is seen to reduce at higher resolution for 5 out of 7 models (Fig. S3e). This difference between the ANOM and AGP results appears plausible when considering how these two indices are defined (Sections 2.3.1 and 2.3.2). While the AGP index identifies blocked situations in terms of the exceedance of fixed thresholds of absolute Z500 (gradients), the ANOM index identifies blocked situations through the exceedance of thresholds defined as quantiles of the model's own large-scale Z500 variability about the model's Z500 mean. In this way, model biases in Z500 mean and variability are partly excluded as a potential source of blocking bias in the ANOM index, and likewise any improvement/deterioration of Z500 mean and variability with resolution will not be fully reflected in an improvement/deterioration in the ANOM blocking index.

The evaluation for the ATL in summer yields similar results to those obtained in winter (Fig. 4). The vast majority of models underestimate blocking, and there is an improvement from CMIP5 to CMIP6. All but one of the PRIMAVERA models are seen to have a smaller mean blocking bias at the higher resolution and with the exception of one model the blocking frequency and spatial correlation also show improvements at higher resolution. As in winter, the sensitivity to resolution is small in the AMIP simulations. The performance of most coupled and corresponding AMIP simulations is similar for most models. The evaluation using the ANOM index (Fig. S4) confirms these results, yet as in winter the sensitivity to resolution is smaller than for the AGP index.

## 4.2  Pacific

The evaluation for the PAC domain and the AGP index (Fig. S5) shows that the domain-mean winter blocking frequency is similar in models and reanalysis, though this is partly due to error compensation within the domain (see Fig. 1). As in the ATL domain, there an improvement is seen from the CMIP5 to the CMIP6 models for all blocking performance metrics. There is no robust improvement with resolution across the PRIMAVERA ensemble, yet 5 of 7 coupled models do show a decrease in RMSE as the resolution is increased. The corresponding evaluation using the ANOM index agrees with these results, i.e. a small improvement from CMIP5 to CMIP6 and no sensitivity of blocking performance to resolution (Fig. S7).

Turning finally to the evaluation for the PAC domain in summer, we find using the AGP index that there is an improvement in simulated blocking in CMIP6 over CMIP5, and there is alo some suggestion of an improvement with resolution in most of the models (Fig. S6). The ANOM index confirms the improvement in CMIP6 over CMIP5. Interestingly, according to this index we also find a clearer improvement at higher resolution than seen with the AGP index (see especially Fig. S8c,e). Referring to the maps of blocking bias shown in Fig. S2b,d, this may be due in part to an improvement at very high latitudes where the AGP index is not defined.

## 5  Blocking persistence

In this section we assess how the persistence of blocking is simulated. This section is organised in a similar way to Sect. 4 with a focus on the ATL domain and the AGP index in the main manuscript (Figures 5 and 6) but results for the PAC domain and using the ANOM index available in the supplement (Figures S9–S14).

## 5.1  Atlantic

The evaluation of winter blocking persistence for the ATL domain and using the AGP index is shown in Fig 5. This figure is organised similarly to the figures in Sect. 4, but here two metrics for blocking persistence, the median persistence of blocking events (top row) and the 90th percentile of blocking persistence (bottom row) are shown.

Models in all of the CMIP5, CMIP6, and PRIMAVERA ensembles tend to underestimate blocking persistence both in coupled and AMIP experiments, by typically 10-15% both for the median and 90th percentile. An improvement towards longer blocking events is seen in the CMIP6 ensemble over CMIP5, though this improvement is not as large, compared to the

ensemble spread, as was found in Sect. 4 in the evaluation of mean blocking frequency (Fig. 3). There is a small increase in the 90th percentile of simulated blocking persistence for most of the PRIMAVERA models as resolution is increased, yet no systematic sensitivity to resolution is seen for the median persistence (for which, also, internal variability is of comparable magnitude to that of resolution sensitivity) nor for the AMIP simulations, as already seen for mean blocking sensitivity to resolution. These findings are corroborated in terms of the corresponding analysis with ANOM index, except for the fact that there is no systematic sensitivity to resolution in blocking persistence when using this index.

For summer (Fig. 4), we see that models both underestimate and overestimate blocking persistence, with a general tendency of no or a small positive bias for the median, and a small negative bias for the 90th percentile, potentially indicating a different shape of the simulated blocking survival functions (see Sect. 2.4), namely a slightly faster decrease with survival time (persistence) in the simulations than in the reanalysis. A small improvement, mainly in the sense of a smaller ensembe spread, can be seen in CMIP6 over CMIP5. There is no systematic sensitivity of simulated blocking persistence to resolution in the PRIMAVERA ensemble. The analyses in terms of the ANOM index are consistent with these results (Fig. S10), noting that here the plots for the median and the 90th percentile are just scaled versions of one another due to the choice of the exponential fit.

We also estimate the respective contributions of biases in the number of events and in persistence to the total blocking bias. To this end, the bias in blocking frequency, i.e. in the total number of blocked days, is decomposed into (i) a component related to the bias in the number of blocking events, (ii) a component related to the bias in persistence, and (iii) a cross term (Fig. S15). We find that the underestimation of the number of events is the main contribution to the total bias, especially in the CMIP5 models, and both in winter and summer. The improvement from CMIP5 to CMIP6 is primarily associated with the fact that more blocking events are simulated in the CMIP6 models. This result is consistent with a similar analysis comparing CMIP5 and CMIP3 models by Davini and D'Andrea (2016).

## 5.2 Pacific

Pacific winter blocking persistence tends to be overestimated by the coupled PRIMAVERA, CMIP5, and CMIP6 models, whereas the PRIMAVERA AMIP simulations scatter around the reanalysis estimate (Fig. S11). There is an improvement from CMIP6 to CMIP5 towards shorter blocking events, but not evidence for sensitivity of simulated blocking persistence to resolution in the PRIMAVERA models. These results are corroborated when using the ANOM index (Fig. S13).

In summer, blocking in the PAC domain is slightly underestimated by most models according to the AGP index (Fig. S12). No sensitivity to resolution is seen for the PRIMAVERA simulations. The spread of the CMIP6 ensemble about the reanalysis estimates is smaller than for CMIP5, which constitutes an improvement. In the case of Pacific summer blocking, the ANOM analysis does not confirm the results obtained with the AGP index (Fig. S14). There is a small overestimation of blocking persistence in most models according to this index, and no systematic difference between the CMIP5 and CMIP6 ensembles. Interestingly, the PRIMAVERA AMIP simulations show a reduction of blocking persistence at higher resolution. This effect is small, but seen in all models and constitutes an improvement for most models. As already argued in Sect. 4, the apparent inconsistency between the AGP and ANOM index may be due to the inclusion of very high-latitude areas in the ANOM

index. Furthermore, the interpretation of the ANOM analysis in particular is complicated as it appears to be affected by error cancellation within the PAC domain with fairly small net results (see also Fig. S2).

For the Pacific, the main feature of the decomposition of the blocking bias is the large spread in the number of simulated blocking events across models (Fig. S15).

## 6 Summary and conclusions

Climate model simulations suffer from long-standing biases in the representation of atmospheric blocking, hampering applications of these simulations in assessing present and future climate impacts associated with blocking such as winter cold spells and summer heat waves. In this study, we revisit the ability of state-of-the-art climate models to represent atmospheric blocking. This analysis is timely due to the recent availability of CMIP6 simulations, including those following the CMIP6-HighResMIP protocol designed to assess the role of model resolution. Our aims are to (i) compare the performance of blocking simulation in CMIP6 and CMIP5 models assessing the net effect of model development between these two generations of multi-model ensembles, and (ii) to assess the sensitivity of simulated blocking to model resolution specifically, using the models/simulations developed in the PRIMAVERA project following the HighResMIP protocol.

Concerning our first aim, we find a clear improvement in simulated blocking in the CMIP6 model ensemble over the CMIP5 ensemble. This improvement is seen robustly for different metrics of mean blocking frequency and blocking persistence, for the Euro-Atlantic and Pacific regions, for winter and summer, and using two different blocking indices (AGP and ANOM) to identify blocking events. The magnitude of the improvement seen depends on the region, season, and blocking index/metric considered — as does the magnitude of the bias itself — yet it is sizeable when compared to the spread of the multi-model ensembles and the total magnitude of the bias. Over a large Euro-Atlantic domain, for example, winter blocking frequency according to the AGP index is seen to be underestimated by 33% for the median CMIP5 model, whereas the same number is 18% for the median CMIP6 model.

We have addressed our second aim using the CMIP6-HighResMIP RIMAVERA simulations to assess the sensitivity of simulated blocking frequency and persistence to resolution. The PRIMAVERA simulations have been designed to assess the role of model resolution specifically by conducting simulations with the same model at both low and high atmosphere resolution (and ocean resolution in coupled setups) without re-tuning the high-resolution version of the model. We find that higher-resolution PRIMAVERA models represent the mean blocking frequency better than the low-resolution models, for the Euro-Atlantic region during winter and summer, and for the Pacific in summer with no sensitivity to resolution seen in winter. This improvement in mean blocking frequency is especially clear for the Euro-Atlantic region and the spatial correlation of the blocking frequency field suggesting that higher-resolution models tend to better simulate the location of blocking occurrences, arguably due to an improvement in the mean circulation over the North Atlantic, which is consistent with previous studies (e.g., Scaife et al., 2011; Zappa et al., 2013; Schiemann et al., 2017; Roberts et al., 2020) and also with the fact that a somewhat larger improvement is seen with the AGP index than with the ANOM index (see discussion in Sect. 4.1). While the improvement with

resolution in the simulated mean blocking frequency is clear, our analysis does not provide robust evidence for a systematic improvement in the simulated blocking persistence.

Our results are consistent with previous findings that the successful simulation of blocking in climate models depends delicately on a range of factors and their interactions, including horizontal and vertical model resolution, orographic boundary conditions, physical parameterisations, and the numerical scheme (Woollings et al., 2018). We corroborate here that horizontal resolution in the atmosphere when increased from, broadly, 100 km to 20 km, is one of these factors and benefits the simulation of blocking frequency. We note that our results regarding model resolution should be considered conservative, as PRIMAVERA models have not been retuned at the higher resolutions. At the same time, we show that an increase in resolution, over the range considered here, will in and of itself not fully remedy blocking biases in models, notably in the persistence of blocking events. We also find that the most recent generation of GCMs continues to be affected by long-standing blocking biases (Fig 1 and 2), albeit at a smaller magnitude in CMIP6 models than in CMIP5 models. This implies that, overall, CMIP6 models strike a better balance of the different factors affecting blocking simulation mentioned above, and that continued model development may further reduce blocking biases.

One question that deserves further attention in future work is the role of the ocean resolution. We find in the PRIMAVERA simulations that the sensitivity to resolution is generally larger in the coupled than in the AMIP simulations. This raises the question if it is not only the better sea-surface temperature mean state that benefits the atmosphere mean state and blocking in higher-resolution models (cf., Scaife et al., 2011) but also the simulation of air-sea-interactions themselves. PRIMAVERA simulations are not designed to answer this question as atmosphere and ocean resolution are increased simultaneously, so that this will have to be addressed by means of future process studies and coordinated experiments increasing ocean and atmosphere resolution separately.

*Data availability.* CMIP5 historical, CMIP6 historical, and PRIMAVERA (through CMIP6-HighResMIP) simulations used in this study are available from the Earth System Grid Federation (ESGF). ERA-40 and ERA-Interim reanalysis data are available from the European Centre for Medium Range Weather Forecasting (ECMWF).

*Author contributions.* RS conducted all of the data analyses and visualisations, and wrote the manuscript. PA, FDR, KJ, MJR, DS, CDR, LT, PLV delivered the PRIMAVERA model simulations. DB and RS developed the ANOM blocking index code. All authors commented on the manuscript.

*Competing interests.* The authors declare that there are no competing interests.

*Acknowledgements.* RS, PA, FDR, KL, MJR, DS, CDR, LT, and PLV acknowledge PRIMAVERA funding received from the European Commission under Grant Agreement 641727 of the Horizon 2020 research programme. Use of the UK NERC (Natural Environment Research Council) CEDA-JASMIN facility is acknowledged for data storage and analysis. DB was supported by the Spanish Government through the PALEOSTRAT (CGL2015-69699-R) and JEDiS (RTI2018-096402-B-I00) projects.

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

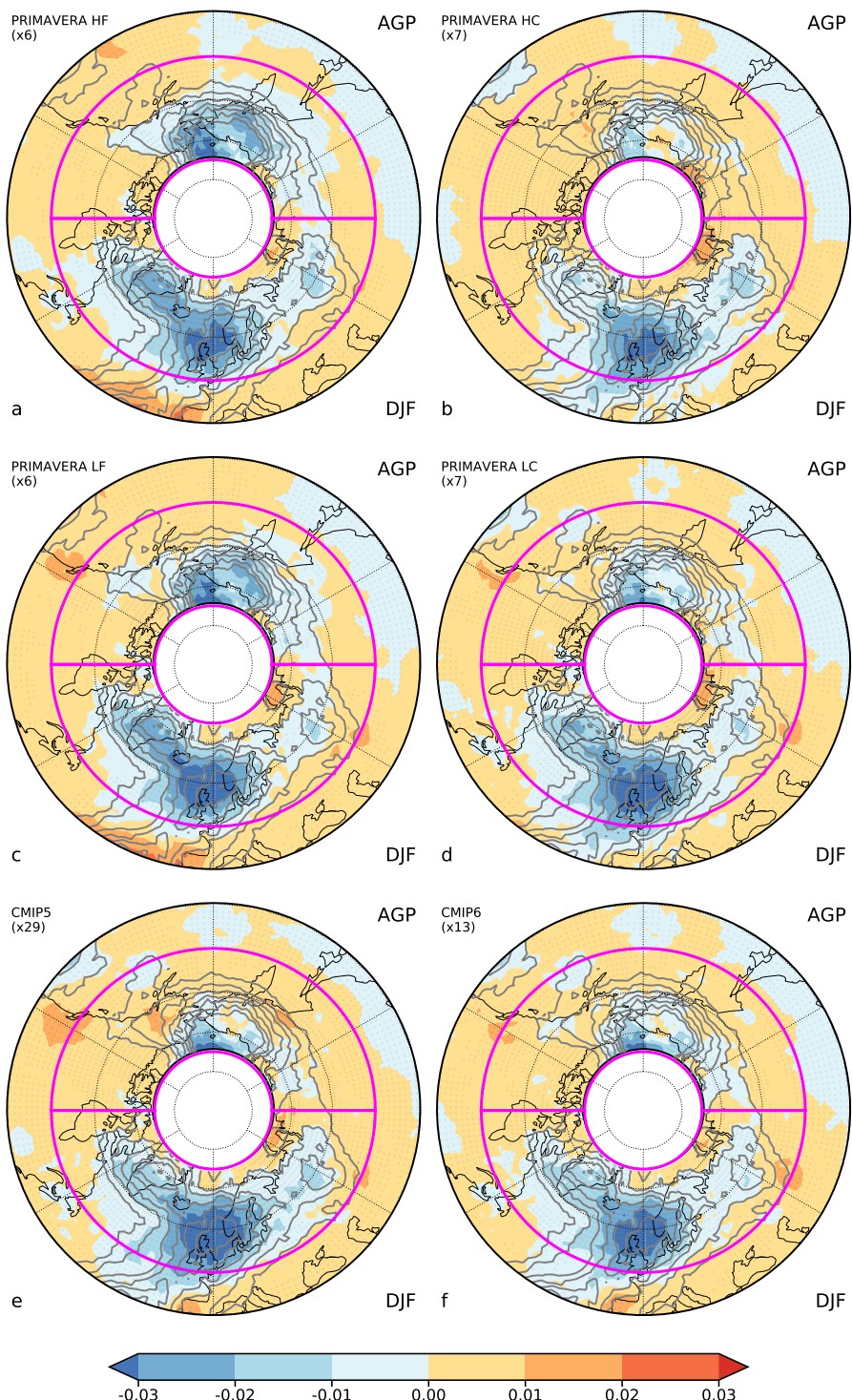

**Figure 1.** Bias in the frequency of blocked days for the AGP index, boreal winter, and (a) high-resolution forced, (b) high-resolution coupled, (c) low-resolution forced, (d) low-resolution coupled PRIMAVERA simulations, and (e) CMIP5, (f) CMIP6 simulations. Stippling shows agreement on the sign of the bias by at least (a,c) 6 of 6, (b,d) 6 of 7, (e) 19 of 29, and (f) 10 of 13 simulations. Grey contour lines show the reanalysis blocking frequency, at contour intervals of 0.01 and starting from 0.01. ATL and PAC evaluation domains are shown by magenta lines.

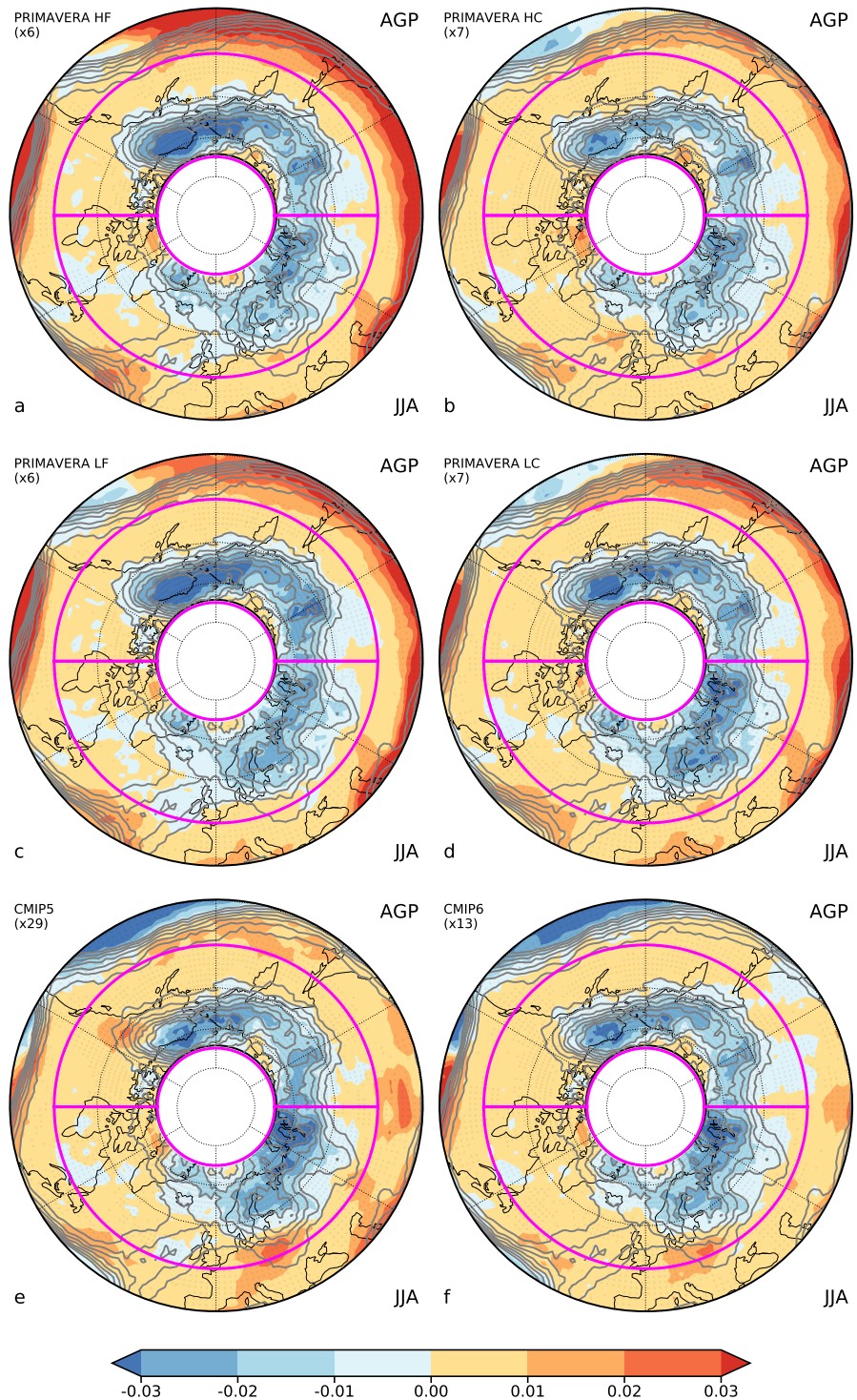

**Figure 2.** As Fig. 1 but for boreal summer. (Blocking frequency bias for the AGP index and (a) high-resolution forced, (b) high-resolution coupled, (c) low-resolution forced, (d) low-resolution coupled PRIMAVERA simulations, and (e) CMIP5, (f) CMIP6 simulations. Stippling shows agreement on the sign of the bias by at least (a,c) 6 of 6, (b,d) 6 of 7, (e) 19 of 29, and (f) 10 of 13 simulations. Grey contour lines show the reanalysis blocking frequency, at contour intervals of 0.01 and starting from 0.01. ATL and PAC evaluation domains are shown by magenta lines.)

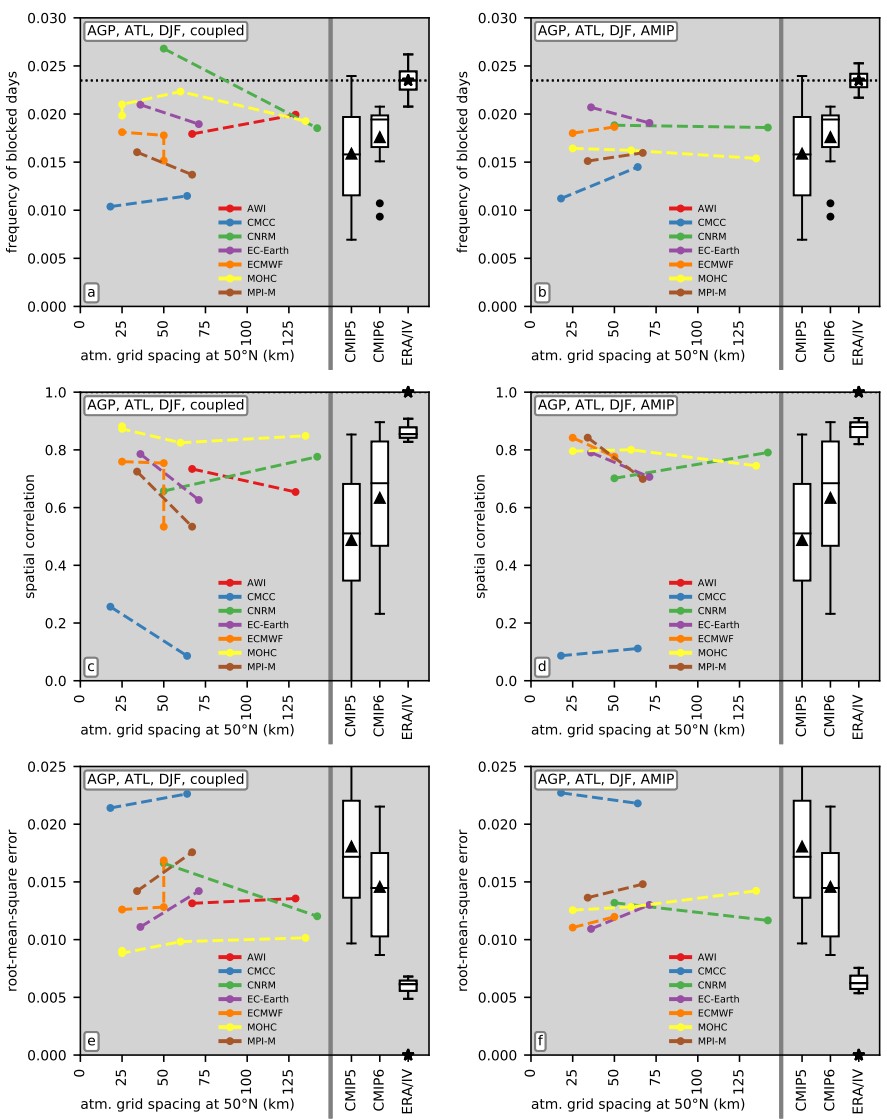

**Figure 3.** Metrics of blocking performance (a,b - blocking frequency, c,d - spatial correlation, e,f - root-mean-square error) for the AGP index and boreal winter, for the ATL domain (-90-90E, 50-75N). The left-hand side of each panel shows metrics for PRIMAVERA simulations at different grid spacings (resolutions). Box-and-whisker plots (boxplots hereinafter) on the right-hand side show distributions of the metric across CMIP5 and CMIP6 simulations in terms of the median, mean (triangle), interquartile range (box, IQR = Q3 - Q1), top whiskers extending to the last datum less than Q3 + 1.5×IQR, and analogously for bottom whiskers. The '*' symbol in the column 'ERA/IV' shows the reanalysis estimate and the boxplot is an estimate of the expected agreement given internal variability (see text).

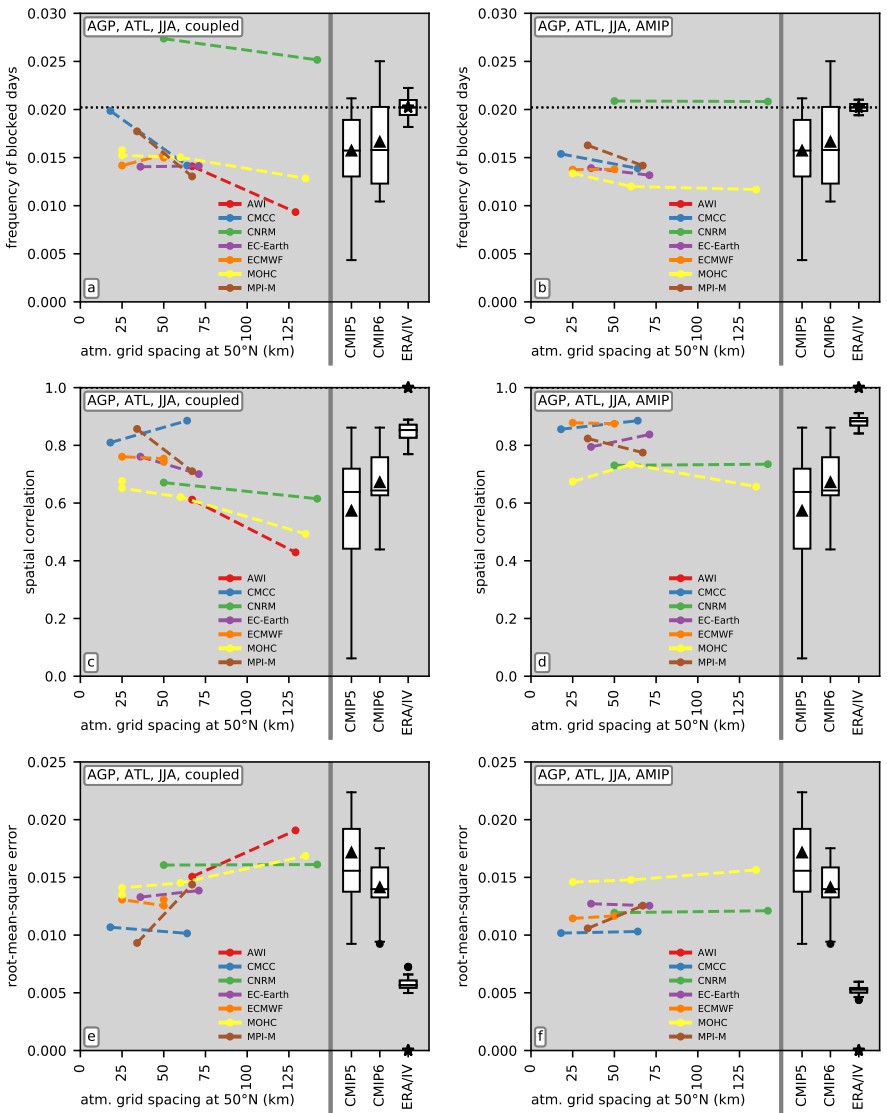

**Figure 4.** As Fig. 3 but for boreal summer. (Metrics of blocking performance (a,b - blocking frequency, c,d - spatial correlation, e,f - root-mean-square error) for the AGP index, for the ATL domain (-90-90E, 50-75N). The left-hand side of each panel shows metrics for PRIMAVERA simulations at different grid spacings (resolutions). Boxplots on the right-hand side show distributions of the metric across CMIP5 and CMIP6 simulations. The '*' symbol in the column 'ERA/IV' shows the reanalysis estimate and the boxplot is an estimate of the expected agreement given internal variability (see text).)

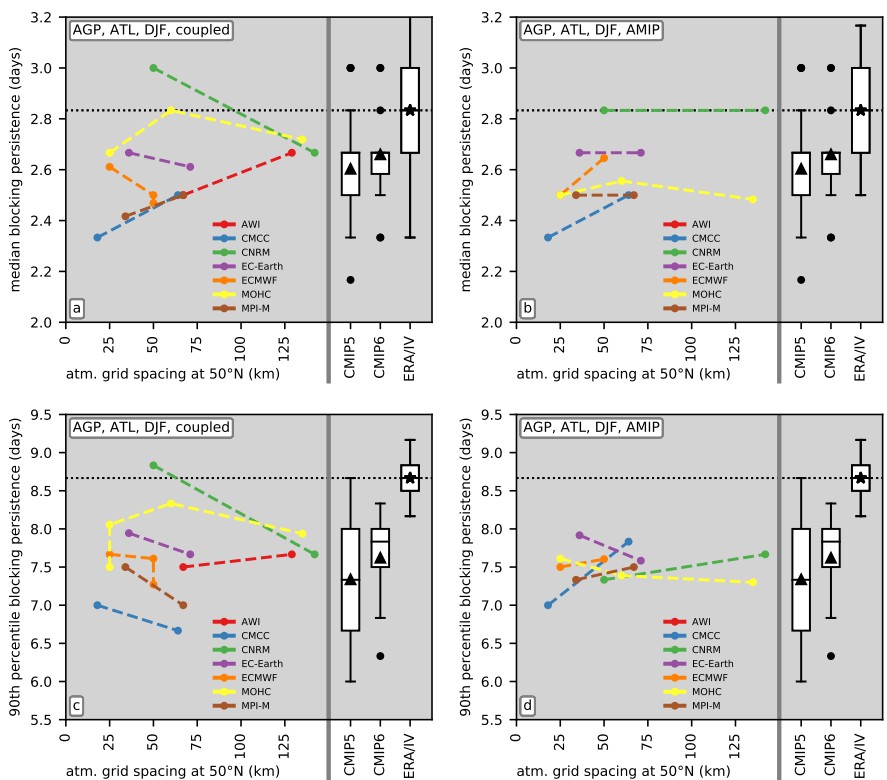

**Figure 5.** Persistence of blocking events (a,b - median, c,d - 90th percentile) for the AGP index and boreal winter, for the ATL domain (-90-90E, 50-75N). The left-hand side of each panel shows metrics for PRIMAVERA simulations at different grid spacings (resolutions). Boxplots on the righ-hand side show distributions of the persistence metric across CMIP5 and CMIP6 simulations. The '*' symbol in the column 'ERA/IV' shows the reanalysis estimate and the boxplot is an estimate of the expected agreement given internal variability (see text).

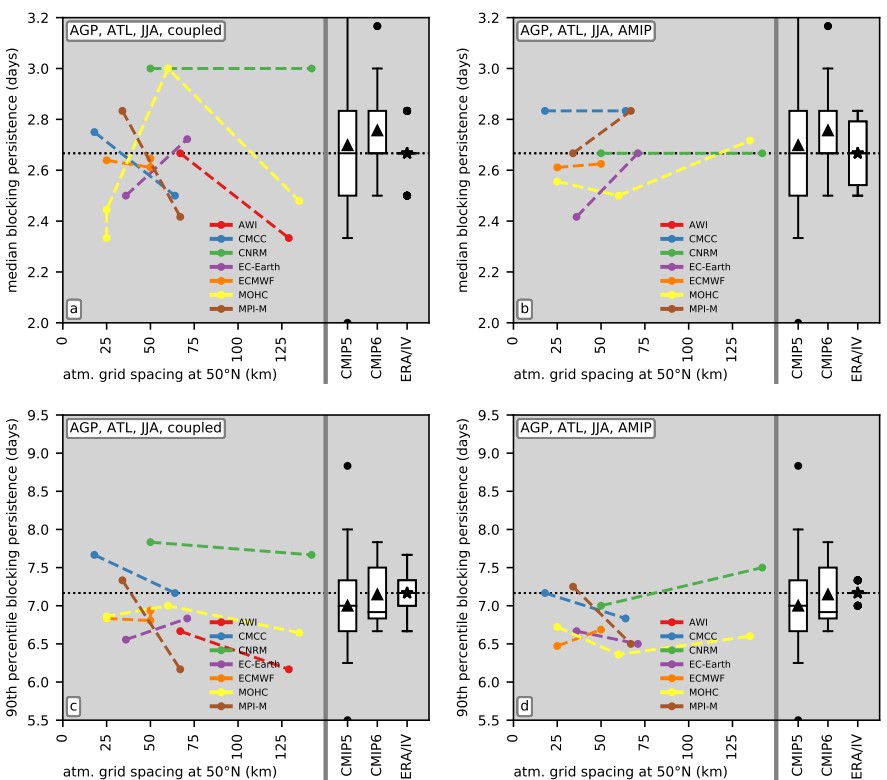

**Figure 6.** As Fig. 5 but for boreal summer. (Persistence of blocking events (a,b - median, c,d - 90th percentile) for the AGP index, for the ATL domain (-90-90E, 50-75N). The left-hand side of each panel shows metrics for PRIMAVERA simulations at different grid spacings (resolutions). Boxplots on the righ-hand side show distributions of the persistence metric across CMIP5 and CMIP6 simulations. The '*' symbol in the column 'ERA/IV' shows the reanalysis estimate and the boxplot is an estimate of the expected agreement given internal variability (see text).)