# Peer review of "Northern Hemisphere blocking simulation in current climate models: evaluating progress from CMIP5 to CMIP6 and sensitivity to resolution"

_Weather and Climate Dynamics, 2019_

## Referee Comment (RC1) · Olivia Romppainen-Martius (Referee) · 13 Feb 2020

This paper compares the representation of Northern Hemisphere blocking in CMIP5 and CMIP6 models for historical periods as well as high-resolution simulations with blocking in reanalysis data. The results show the CMIP6 data capture blocking better than CMIP5 and that higher resolution contributes to a better representation of blocks. However, simply increasing resolution does not completely remove biases. The results are relevant and the paper is well organized and written and the figures are clear. I therefore recommend publication after minor revisions.

Minor points: 1) Suggest to replace current with a more specific description in the title 2) L28 List the references here already 3) L69 overviewed → listed 4) L110ff How

can this blocking indicator be affected by temperature trends? 5) L127 Why do you include shorter lived anomalies as well? 6) L146 Mention by how much they are underestimated 7) L299ff Can you further clarify this statement, it is not obvious to me why this is the case. 8) A general point: please add a discussion on whether the underestimation of the longevity is the explanation for the frequency biases or if there are also differences in the number of events? 9) Figure 3ff: I do not understand why the reanalysis estimate is outside of the reanalysis variability box plot.

---

## Referee Comment (RC2) · Anonymous Referee #2 · 25 Feb 2020

Review of "The representation of Northern Hemisphere blocking in current global climate models" by Schiemann et al.

This paper examines the current state-of-the-art, and the most recent prior state-of-the-art, in blocking simulation by climate models. The study is well designed, examining both CMIP ensembles and complementary experiments by PRIMAVERA models to test the impact of model resolution changes. It also usefully tests the robustness of the results to the choice of blocking index. The paper is very well written, well organized, and concise. I'm not sure if my comments & questions should be considered major or minor revisions. At any rate I think it should be very feasible for the authors to address these and I hope to see this paper published.

Main comments:

[Figure]

1. Overall I'm surprised by how little change there is in blocking biases from CMIP5 to CMIP6. The small size of the currently available CMIP6 ensemble (13 models) is a concern (although there's not much the authors can do about this). But it seems to me that a main conclusion from this study could be that the pervasive blocking biases documented in previous generations of models are more or less unchanged in these new ones. The authors do note a sizeable change in the median DJF Atlantic blocking frequency bias, but overall most of the models seem to look the same. However, having only 13 models for CMIP6 could mean that this (or other) changes aren't robust. Perhaps a test could be to see what spread in CMIP5 results is obtained by drawing random samples of 13 models from the CMIP5 ensemble - do the CMIP6 models consistently beat these?

2. Only a single ensemble member is used for each CMIP model, and a number of the PRIMAVERA experiments have one member. Are these samples big enough for robust results? More ensemble members are available for many of the CMIP models. On the line plots for PRIMAVERA models (left panel of each plot in Figs 3-6) could the uncertainty in each metric be shown? Perhaps a bootstrap test would work, randomly sampling years from each experiment to get a confidence interval for the metric. Many of the changes with resolution seen in these plots are marginal or differ widely among the models. These could represent genuine inter-model differences, but also sampling variability. Re. the boxplots of CMIP models (Figs 3-6), there are many cases where it looks like the CMIP6 distribution could be drawn randomly from the CMIP5. Hence it would be useful to do significance tests of the mean changes for these metrics.

3. For PRIMAVERA models, it's interesting that in many cases the AMIP runs show little or no change with resolution but the coupled runs do. One interpretation is that the mean state doesn't change as much with resolution in the AMIP runs because the SSTs are fixed, whereas they can change in the coupled runs. Can this be tested by looking at the mean states, e.g. by adding a mean-state metric to the boxplots? Perhaps RMSE of time-mean Z500 or similar.

Comments & suggestions by line number:

8: "and in" –> "during"

62: Should state if "high-resolution" refers to changing just horizontal resolution, as Table 1 seems to suggest, or if some of the PRIMAVERA models also increase their vertical resolution.

75: I'm surprised one historical simulation is enough for looking at persistence, given that some long-duration blocking events will be quite rare occurrences. Multiple historical ensemble members are available for many CMIP5 models and most CMIP6 models (some modelling centres have even submitted "large ensembles" of these). These extra members could be used to get more robust results, or at least to test if one ensemble member per model is sufficient.

Somewhere in Sec. 2.1: it would help to comment what are the ranges of resolution spanned by the CMIP5 and CMIP6 ensembles. Are the changes in blocking frequency between CMIP5 and CMIP6 roughly what would be expected, based on the PRIMAVERA analysis, given the change in resolution from CMIP5 to CMIP6?

115: "throughout" –> "at each gridpoint throughout" (if this is what you mean)

117: "Z500 anomaly" –> "Z500 anomaly at each gridpoint" (again, I presume this is what you mean)

148: In the Pacific high latitudes in some cases (in Fig 1) there's not much stippling, so this seems to be one region where models don't share the same bias.

150: My initial reaction to panels (e) & (f) of Fig 1 was that there's essentially no difference in DJF blocking frequency between CMIP5 & CMIP6, for both Atlantic and Pacific. Looking more closely I do see that CMIP5 has a larger area of negative bias in the Atlantic basin, but it's not a very big difference. It seems hard to believe this is a reduction of 1/3 in the bias, as the abstract says. Is that because the change in median (referred to in the abstract) is a lot more dramatic than the change in mean

(shown in Fig 1)? Fig 3, top left, seems to show a bigger shift in the median than the mean between CMIP5 and CMIP6, assuming the triangle is the mean. Perhaps it's worth plotting the difference in ensemble-mean blocking frequency between CMIP5 and CMIP6, and calculating its statistical significance.

151: "Fig. 2e,f" –> "Fig. 1e,f"

152: As with the CMIPs, I don't see very big differences between the different PRI-MAVERA cases, so plotting differences & their significances could help. Given there aren't too many PRIMAVERA models, robustness could be a concern. However there is another issue with the H,L groups used to split up the PRIMAVERA models: in some cases the "low" resolution of one model group is similar to the "high" of another model group - e.g. the low-res CMCC models (64 km atm. grid) compared to the high-res AWI model (67 km atm. grid). So these composite groups, as plotted in Figs 1 & 2, are mixing together models of different resolutions. If changes in blocking frequency with resolution follow a similar pattern in all the models, then maybe this is ok. However Figs 3-6 suggest the models can differ widely in how blocking changes with resolution. What do the plots look like if instead models were grouped just based on their horizontal grid spacings?

157: It's hard to see much difference between CMIP5 and CMIP6 for summer blocking. The changes over the Baltic and Siberia are so small I wonder if they're just sampling variations. As with DJF, it would be useful to plot the difference & its statistical significance.

165: For DJF I see some differences in the Pacific during DJF among the PRIMAVERA models.

182: This could also be done with the HadGEM3 and EC-EARTH groups, is a similar uncertainty range obtained?

200: "mean and variability" –> "mean" (shouldn't ANOM focus on the variability?)

[Figure]

204: "underestimates" –> "underestimate"

236: "interval" –> "internal"

236: "comparatively large": as compared to the median?

240: "Moving on to summer (Fig. 4)" –> "For summer (Fig. 6)"

242: Refer to Sec. 2.4 to remind reader what a "blocking survival function" is. Also 2nd instance of "survival" misspelled.

243: I'm not sure that smaller ensemble spread indicates an improvement, given the CMIP6 models number less than half (13) than the CMIP5 models (29).

251: In Fig. S13, CMIP5 & CMIP6 look essentially the same to me.

254: "and improvement" –> "an improvement"

254: Again, I'm not sure that smaller ensemble spread for a smaller ensemble indicates improvement. Can this be quantified somehow - perhaps by drawing random groups of 13 models from the CMIP5 ensemble and seeing how likely it is to get a spread the size of CMIP6's by chance?

263: insert comma after "blocking"

271: Although blocking frequency has improved in CMIP6, given how similar the CMIPs look in Figs 1 & 2 I think calling it a "very clear" improvement may overstate the change.

271: "blocking" –> "blocking frequency" (since blocking persistence shows little change)

287: This sounds plausible, but is it actually happening in the PRIMAVERA models? Given that they aren't retuned at the higher resolution, it's possible that the opposite is true, that their mean states are actually worse at the higher resolution. This could be checked.

294: "blocking" –> "blocking frequency"

295: "increase in resolution in and of itself" –> "increase in resolution over the range considered here in and of itself"

300: This last sentence is kind of long, and speculative. Perhaps instead just say the results suggest that further model development is expected to reduce blocking biases.

304: I agree this is possible, but could a simpler interpretation might be: coupled models show larger sensitivity to resolution that AMIP models because the SST mean state improves with resolution in the coupled models (whereas it can't in the AMIP ones, because it's prescribed). This suggests the blocking changes with resolution are more to do with mean-state changes than changes in the variability. Do you think the results support this conclusion?

Figs 1 & 2: perhaps change contour levels so that zero isn't shown, e.g. begin at +/- 0.005. It might help reduce the appearance of noisiness in the plots.

Figs 3-6: define the different elements of the boxplots (is "box-whisker" plot a more typical name for these?). What is the triangle, quantiles for the box, the circles, etc. Panel labels (a,b,...) could be useful.

Captions in general: since so many of the plots are of the same type, it would be useful for just the first plot of each kind to have a full descriptive caption, and then for subsequent plots just say, "As Fig. X, but for [JJA, ANOM, etc]". Otherwise the reader has to go through each caption to find out if much has changed from the previous plot.

---

## Author Comment (AC1) · 8 Apr 2020

**Author response to referee comments on manuscript wcd-2019-19: "The representation of Northern Hemisphere blocking in current global climate models" by Reinhard Schiemann, Panos Athanasiadis, David Barriopedro,**

Francisco Doblas-Reyes, Katja Lohmann, Malcolm J. Roberts, Dmitry Sein, Christopher D. Roberts, Laurent Terray, and Pier Luigi Vidale

**Referee comment 1, by Olivia Romppainen-Martius**

This paper compares the representation of Northern Hemisphere blocking in CMIP5 and CMIP6 models for historical periods as well as high-resolution simulations with blocking in reanalysis data. The results show the CMIP6 data capture blocking better than CMIP5 and that higher resolution contributes to a better representation of blocks. However, simply increasing resolution does not completely remove biases. The results are relevant and the paper is well organized and written and the figures are clear. I therefore recommend publication after minor revisions.

We thank the referee for her interesting comments. Our point-by-point responses follow below, in blue, with the original referee comments shown in black. Modifications to the manuscript are shown in orange.

**Minor points**

**1) Suggest to replace current with a more specific description in the title**

We replace the title with the following: "Northern Hemisphere Blocking simulation in current climate models: evaluating progress from CMIP5 to CMIP6 and sensitivity to resolution".

**2) L28 List the references here already**

Done.

3) L69 overviewed -> listed

Done.

**4) L110ff How can this blocking indicator be affected by temperature trends?**

The blocking indices used in this study are calculated from the geopotential height at 500 hPa (Z500). Z500 depends on the temperature of the atmospheric layer below 500 hPa and is therefore affected by temperature trends, for example in response to anthropogenic forcing (Christidis & Stott, 2015). Thus, trends in blocking evaluated with these indices will comprise both thermodynamic and dynamic components. This issue can be dealt with by choosing indices defined in terms of a dynamical variable and by defining climatologies/thresholds separately, say, for a current and future climate period (e.g., Schwierz et al., 2004; Sillmann & Croci-Maspoli, 2009).

For the purpose of this study, namely model evaluation in a historical period, these issues are less critical, and we choose to work with Z500 because it is readily available for the different multi-model ensembles evaluated.

**5) L127 Why do you include shorter lived anomalies as well?**

It appears natural to us to obtain results for the empirical survival function through the entire range of persistence times. We do, however, add the subclause "..., which are not strictly considered to be blocks," in Line 129 in response to the referee's comment.

**6) L146 Mention by how much they are underestimated**

Done. See also comment 11 by referee 2.

**7) L299ff Can you further clarify this statement, it is not obvious to me why this is the case**

We have reformulated, also in response to the related comment 33 by referee 2, as follows (Line 299): "... mentioned above, and that continued model development may further reduce blocking biases."

**8) A general point: please add a discussion on whether the underestimation of the longevity is the explanation for the frequency biases or if there are also differences in the number of events?**

We have analysed this by decomposing the bias in blocking frequency, i.e. in the total number of blocked days, into (i) a component related to the bias in persistence, (ii) a component related to the bias in the number of blocking events, and (iii) a cross term that is small in most cases. We find that the underestimation of the number of events is the main contribution to the total bias, especially in the CMIP5 models. The improvement seen from CMIP5 to CMIP6 is primarily associated with the fact that more blocking events are simulated in the CMIP6 models. We have added a brief discussion of these results to Section 5 of the manuscript. Our results are consistent with a similar analysis carried out by Davini & D'Andrea, 2016, who find that the blocking frequency change between CMIP5/AMIP5 and multi-model ensembles from earlier MIPs is dominated by an increase in the simulated number of blocking events.

**9) Figure 3ff: I do not understand why the reanalysis estimate is outside of the reanalysis variability box plot**

This is because the correlation and root-mean-square error (RMSE) metrics are bounded. We estimate internal variability by forming pairs from an ensemble of simulations with a single model, and then calculate the correlations (RMSEs) between the blocking frequency patterns for each of these pairs. The expected value of these correlations (RMSEs) is less than 1 (greater then 0) due to internal variability as shown by the boxplots in the 'ERA/IV' column of the plots. The reanalysis estimates are simply shown at correlation=1 (RMSE=0) and do not take into account internal variability.

**References**

- Christidis, N., & Stott, P. A. (2015). Changes in the geopotential height at 500 hPa under the influence of external climatic forcings. *Geophysical Research Letters*, 42(24), 10798–10806. https://doi.org/10.1002/2015GL066669
- Davini, P., & D'Andrea, F. (2016). Northern Hemisphere Atmospheric Blocking Representation in Global Climate Models: Twenty Years of Improvements? *Journal of Climate*, *29*(24), 8823–8840. https://doi.org/10.1175/JCLI-D-16-0242.1
- Schwierz, C., Croci-Maspoli, M., & Davies, H. C. (2004). Perspicacious indicators of atmospheric blocking. *Geophysical Research Letters*, *31*(6), L06125. https://doi.org/10.1029/2003GL019341
- Sillmann, J., & Croci-Maspoli, M. (2009). Present and future atmospheric blocking and its impact on European mean and extreme climate. *Geophysical Research Letters*, *36*(10), 1–6. https://doi.org/10.1029/2009GL038259

---

## Author Comment (AC2) · 8 Apr 2020

Author response to referee comments on manuscript wcd-2019-19:
"The representation of Northern Hemisphere blocking in current global climate models"
by
Reinhard Schiemann, Panos Athanasiadis, David Barriopedro,
Francisco Doblas-Reyes, Katja Lohmann, Malcolm J. Roberts, Dmitry Sein,
Christopher D. Roberts, Laurent Terray, and Pier Luigi Vidale

Referee comment 2 (anonymous)

This paper examines the current state-of-the-art, and the most recent prior state-of-the-art, in blocking simulation by climate models. The study is well designed, examining both CMIP ensembles and complementary experiments by PRIMAVERA models to test the impact of model resolution changes. It also usefully tests the robustness of the results to the choice of blocking index. The paper is very well written, well organized, and concise. I'm not sure if my comments & questions should be considered major or minor revisions. At any rate I think it should be very feasible for the authors to address these and I hope to see this paper published.

We thank the referee for their overall positive review. Our point-by-point responses follow below, in blue, with the original referee comments shown in black. Modifications to the manuscript are shown in orange. Cross-references with simple numerals (Table 1, Figure 1, …) refer to the originally submitted manuscript, labels S1, S2, … refer to figures in the originally submitted supplement, and labels AR2-1, … refer to this author response.

Main comments:

1. Overall I'm surprised by how little change there is in blocking biases from CMIP5 to CMIP6. The small size of the currently available CMIP6 ensemble (13 models) is a concern (although there's not much the authors can do about this). But it seems to me that a main conclusion from this study could be that the pervasive blocking biases documented in previous generations of models are more or less unchanged in these new ones. The authors do note a sizeable change in the median DJF Atlantic blocking frequency bias, but overall most of the models seem to look the same. However, having only 13 models for CMIP6 could mean that this (or other) changes aren't robust. Perhaps a test could be to see what spread in CMIP5 results is obtained by drawing random samples of 13 models from the CMIP5 ensemble - do the CMIP6 models consistently beat these?

This is an interesting suggestion and we have tested this. Resampling results for the ATL domain in winter are shown in Figure AR2-1, showing that the improvement from CMIP5 to CMIP6 is robust, even for the ensemble means of the different metrics.

2a. Only a single ensemble member is used for each CMIP model, and a number of the PRIMAVERA experiments have one member. Are these samples big enough for robust results? More ensemble members are available for many of the CMIP models. On the line plots for PRIMAVERA models (left panel of each plot in Figs 3-6) could the uncertainty in each metric be shown? Perhaps a bootstrap test would work, randomly sampling years from each experiment to get a confidence interval for the metric. Many of the changes with resolution seen in these plots are marginal or differ widely among the models. These could represent genuine inter-model differences, but also sampling variability.

We use multi-model ensembles each member of which is a simulation of about 60 years in length. For the CMIP5 and CMIP6 ensembles, we use 29 and 13 such simulations, respectively, and we use several ensemble members for the PRIMAVERA models where available. The objectives of our study are to assess (i) to what extent the increase in horizontal resolution benefits the simulation of blocking in

the PRIMAVERA ensemble and (ii) how model performance in simulating blocking has evolved from CMIP5 to CMIP6. The simulation data we use are suitable to meet these objectives. We have estimated internal (sampling) variability in the different blocking metrics from a 6-member ensemble of ECMWF-IFS simulations (Figures 3-6, column 'ERA/IV') showing that internal variability is much smaller than inter-model spread. We can therefore meet our first objective by comparing the CMIP5 and CMIP6 ensemble distributions directly (Figures 3-6), and the difference between these ensembles is now also illustrated by the analysis in response to comment 1. We prefer to estimate internal variability from an ensemble of simulations as this reflects variability across timescales.

[Figure]

*Figure AR2-1. Metrics of blocking performance for the AGP index and boreal winter, for the ATL domain. Boxplot statistics on the left are for the full CMIP5 (29 members) and CMIP6 (13 members) ensembles, as in Figure 3 of the main manuscript. The 10 remaining boxplots are for random 13-member subsamples of the CMIP5 ensemble. The horizontal dashed and dotted lines show the CMIP6 ensemble median and mean, respectively.*

Regarding our second objective, it is true that the resolution sensitivity is not always larger than internal variability for individual models/metrics but it is the behaviour of the entire ensemble we are interested in. For example, Figure 3c shows that 6 out of 7 models represent the winter ATL blocking pattern better at the higher resolution. On its own, this represents only mild evidence (a p-value of 0.125 results from working out the binomial probabilities) but this needs to be seen in context of a number of previous studies that have shown an improvement in simulated blocking with resolution,

and of the general fact that an improved numerical solution at higher resolution is perfectly plausible. To make clearer that there is some variation across models, we have added "in most of the [models]" in line 12 of the abstract.

2b. Re. the boxplots of CMIP models (Figs 3-6), there are many cases where it looks like the CMIP6 distribution could be drawn randomly from the CMIP5. Hence it would be useful to do significance tests of the mean changes for these metrics.

This question has been addressed in the response to comment 1, the CMIP6 distribution cannot be randomly drawn from the CMIP5 distribution.

3. For PRIMAVERA models, it's interesting that in many cases the AMIP runs show little or no change with resolution but the coupled runs do. One interpretation is that the mean state doesn't change as much with resolution in the AMIP runs because the SSTs are fixed, whereas they can change in the coupled runs. Can this be tested by looking at the mean states, e.g. by adding a mean-state metric to the boxplots? Perhaps RMSE of time-mean Z500 or similar.

Yes, this is plausible and several authors have shown that mean state and SST biases are closely related to blocking biases (Scaife et al., 2011; Schiemann et al., 2017). The relationship between biases in SST, the mean circulation (jet stream, storm tracks), and blocking is further explored in a separate study that is in preparation by PRIMAVERA authors (Athanasiadis et al.) and is beyond the scope of this paper.

4. 8: "and in" –> "during"

Done.

5. 62: Should state if "high-resolution" refers to changing just horizontal resolution, as Table 1 seems to suggest, or if some of the PRIMAVERA models also increase their vertical resolution.

Added "horizontal".

6. 75: I'm surprised one historical simulation is enough for looking at persistence, given that some long-duration blocking events will be quite rare occurrences. Multiple historical ensemble members are available for many CMIP5 models and most CMIP6 models (some modelling centres have even submitted "large ensembles" of these). These extra members could be used to get more robust results, or at least to test if one ensemble member per model is sufficient.

The same arguments apply here as in response to comments 1 and 2, and a resampling experiment as in response to comment 1 shows that CMIP5 and CMIP6 are significantly different also for the persistence metrics (not shown).

7. Somewhere in Sec. 2.1: it would help to comment what are the ranges of resolution spanned by the CMIP5 and CMIP6 ensembles. Are the changes in blocking frequency between CMIP5 and CMIP6 roughly what would be expected, based on the PRIMAVERA analysis, given the change in resolution from CMIP5 to CMIP6?

The CMIP5 and CMIP6 model resolutions correspond to grid spacings between about 100 and 400 km, and 100 and 300 km, respectively. About 200 km can be considered typical for both ensembles and the actual increase in resolution between the two is small. PRIMAVERA is the European contribution to CMIP6-HighResMIP, the dedicated CMIP6 high-resolution modelling effort that explores a different

range of resolutions (about 20 to 100 km). The referee's question can therefore not be meaningfully answered, but we mention the range of CMIP5 and CMIP6 resolutions in Section 2.1.

8. 115: "throughout" –> "at each gridpoint throughout" (if this is what you mean)

The threshold does not vary spatially for this index. We have added "spatially invariable" to line 114 to make this clearer.

9. 117: "Z500 anomaly" –> "Z500 anomaly at each gridpoint" (again, I presume this is what you mean)

See previous comment.

10. 148: In the Pacific high latitudes in some cases (in Fig 1) there's not much stippling, so this seems to be one region where models don't share the same bias.

There is stippling almost everywhere where the observed blocking frequency is high, including in the northern part of the region of high observed blocking occurrence in the Pacific. We therefore stand by our statement that blocking biases are pervasive in models. At the same time our plot allows for detailed regional inspection, showing, for example, small biases (and therefore little agreement on the sign of these biases) in parts of the North Pacific.

11. 150: My initial reaction to panels (e) & (f) of Fig 1 was that there's essentially no difference in DJF blocking frequency between CMIP5 & CMIP6, for both Atlantic and Pacific. Looking more closely I do see that CMIP5 has a larger area of negative bias in the Atlantic basin, but it's not a very big difference. It seems hard to believe this is a reduction of 1/3 in the bias, as the abstract says. Is that because the change in median (referred to in the abstract) is a lot more dramatic than the change in mean (shown in Fig 1)? Fig 3, top left, seems to show a bigger shift in the median than the mean between CMIP5 and CMIP6, assuming the triangle is the mean. Perhaps it's worth plotting the difference in ensemble-mean blocking frequency between CMIP5 and CMIP6, and calculating its statistical significance.

We agree with the referee that it can be difficult to estimate spatially averaged differences from a map, and we therefore show both maps (Figures 1 and 2) as well as three metrics aggregated over large domains (Figures 3 and 4). Both ensemble medians and means are shown for CMIP5 and CMIP6, specifically in Figure 3a for the example discussed here. We reproduce these numbers in Table AR2-1. For this example, the difference in the mean is smaller than that for the median as it is not robust to the two outliers shown. For the spatial correlation and root-mean-square error (Figure 3c,e), however, the differences between mean and median is small. The robustness of the difference between the CMIP5 and CMIP6 ensembles was discussed in response to comment 1.

*Table AR2-1. Median and mean blocking frequency for the CMIP5 and CMIP6 ensembles in the ATL domain, for the AGP blocking index during winter. Percent underestimation with respect to the reanalysis estimate is also shown. The reanalysis estimate is 0.0235.*

|                     | CMIP5  | CMIP6  |
|---------------------|--------|--------|
| median              | 0.0158 | 0.0194 |
| 1 - (median / ERA)  | 32.7%  | 17.3%  |
|                     |        |        |
| mean                | 0.0159 | 0.0176 |
| 1 - (mean / ERA)    | 32.4%  | 25.1%  |

We thank the referee for bringing up this example as it made us realise that the percentages reported in the abstract were a little too approximate (obtained by just reading off Figure 3a). Having done the full precision calculation, we correct the numbers reported in the abstract and Lines 277,278 to 33% and 18%. (Rounding up and in the same direction appears reasonable to us for reporting these biases.)

12. 151: "Fig. 2e,f" –> "Fig. 1e,f"

Corrected, thank you.

13. 152: As with the CMIPs, I don't see very big differences between the different PRIMAVERA cases, so plotting differences & their significances could help. Given there aren't too many PRIMAVERA models, robustness could be a concern. However there is another issue with the H,L groups used to split up the PRIMAVERA models: in some cases the "low" resolution of one model group is similar to the "high" of another model group - e.g. the low-res CMCC models (64 km atm. grid) compared to the high-res AWI model (67 km atm. grid). So these composite groups, as plotted in Figs 1 & 2, are mixing together models of different resolutions. If changes in blocking frequency with resolution follow a similar pattern in all the models, then maybe this is ok. However Figs 3-6 suggest the models can differ widely in how blocking changes with resolution. What do the plots look like if instead models were grouped just based on their horizontal grid spacings?

After similar considerations when preparing the manuscript, we decided to present maps for the four PRIMAVERA sub-ensembles (Table 1, Figures 1 and 2) but to then also show domain-averaged metrics for individual models (Figures 3 and 4), plotted against the resolution of these individual models. This presents results in way that is suitable to meet the two objectives of our study, and it strikes a balance between (i) showing spatially resolved information, (ii) showing variability between different models, and (iii) keeping the total number of plots manageable.
We agree with reviewer that there is some variation in the resolution sensitivity across the PRIMAVERA models – we have discussed this in the response to comment 2 and made a change in the abstract.

14. 157: It's hard to see much difference between CMIP5 and CMIP6 for summer blocking. The changes over the Baltic and Siberia are so small I wonder if they're just sampling variations. As with DJF, it would be useful to plot the difference & its statistical significance.

The manuscript states that these differences are small, nonetheless our results show an improvement from CMIP5 to CMIP6 also for summer (Figure 4).

15. 165: For DJF I see some differences in the Pacific during DJF among the PRIMAVERA models.

We have reformulated "and these improvements are not seen for the Pacific" to be clearer.

16. 182: This could also be done with the HadGEM3 and EC-EARTH groups, is a similar uncertainty range obtained?

Yes, a similar uncertainty range is obtained (Figure AR2-2), and the choice of the model does not alter any of the conclusions of our study.

[Figure]

*Figure AR2-2. Internal variability (AGP, DJF, winter) estimated for three different models (ECMWF-IFS-HR – 6 members, HadGEM3-GC31-LL – 8 members, and EC-Earth3P-HR – 3 members). Compare Figure 3a,c,e. Individual values are shown for EC-Earth3P-HR as only three pairs can be formed from the three simulations.*

17. 200: "mean and variability" –> "mean" (shouldn't ANOM focus on the variability?)

Mean and variability is correct here, as it refers to the preceding sentence ("…, the ANOM index identifies blocked situations through the exceedance of thresholds defined as quantiles of the model's own large-scale variability about the model's Z500 mean.").

18. 204: "underestimates" –> "underestimate"

Corrected.

19. 236: "interval" –> "internal"

Corrected.

20. 236: "comparatively large": as compared to the median?

As compared to resolution sensitivity of persistence. We have reformulated ("(for which, also, internal variability is of comparable magnitude to that of resolution sensitivity)").

21. 240: "Moving on to summer (Fig. 4)" –> "For summer (Fig. 6)"

Corrected.

22. 242: Refer to Sec. 2.4 to remind reader what a "blocking survival function" is. Also 2$^{nd}$ instance of "survival" misspelled.

Done and corrected.

23. 243: I'm not sure that smaller ensemble spread indicates an improvement, given the CMIP6 models number less than half (13) than the CMIP5 models (29).

The boxplots do show that there are fewer models with very large persistence biases in CMIP6 than in CMIP5 for both the AGP index (Figure 6) and the ANOM index (Figure S10). Calling this a small improvement appears justified to us.

24. 251: In Fig. S13, CMIP5 & CMIP6 look essentially the same to me.

The ensemble spread is smaller than that in CMIP5 and the median is closer to the reanalysis estimate for both AMIP and coupled simulations, and for both persistence metrics (Figure S13). While we have discussed the fact that bias reductions seen in the ANOM index are generally smaller than those seen for the AGP index (Line 196ff.), these results support the AGP results (Figure S11) of an improved simulation of persistence in CMIP6.

25. 254: "and improvement" –> "an improvement"

Corrected.

26. 254: Again, I'm not sure that smaller ensemble spread for a smaller ensemble indicates improvement. Can this be quantified somehow - perhaps by drawing random groups of 13 models from the CMIP5 ensemble and seeing how likely it is to get a spread the size of CMIP6's by chance?

Even if a somewhat larger CMIP6 ensemble would be preferable in principle, the boxplot statistics comprise quantiles (and the mean) and do not directly depend on sample size. The issue of robustness has been discussed above (see, e.g., comment 6).

27. 263: insert comma after "blocking"

Corrected.

28. 271: Although blocking frequency has improved in CMIP6, given how similar the CMIPs look in Figs 1 & 2 I think calling it a "very clear" improvement may overstate the change.

Jointly with the three different domain-aggregated metrics (Figures 3 and 4) the evidence is clear, see also above discussions about robustness, e.g. comment 1. The magnitude of the improvement may not be very large compared to the bias in each case, but this is quantified in our plots. With all that, the formulation "very clear" is unnecessary and we have changed this to just say "clear".

29. 271: "blocking" –> "blocking frequency" (since blocking persistence shows little change)

The CMIP5 to CMIP6 improvement is also seen in the persistence as discussed above (comments 6, 24, 26).

30. 287: This sounds plausible, but is it actually happening in the PRIMAVERA models? Given that they aren't retuned at the higher resolution, it's possible that the opposite is true, that their mean states are actually worse at the higher resolution. This could be checked.

The mean state is known to be important for simulated blocking, and this argument is also consistent with the fact that we see larger improvements in the AGP than in the ANOM index. We have modified this sentence adding two more references (Scaife et al., 2011; Schiemann et al., 2017) and making this additional argument. Our use of the word "arguably" indicates that this is not a result of our study but points to possible future work. Indeed, as discussed in response to comment 3, a separate study addressing this issue is in preparation by PRIMAVERA authors.

31. 294: "blocking" –> "blocking frequency"

Corrected.

32. 295: "increase in resolution in and of itself" –> "increase in resolution over the range considered here in and of itself"

Done.

33. 300: This last sentence is kind of long, and speculative. Perhaps instead just say the results suggest that further model development is expected to reduce blocking biases.

We agree that this sentence is speculative and adds little to the paper. We have reformulated as follows (Line 299): "… mentioned above, and that continued model development may further reduce blocking biases."

34. 304: I agree this is possible, but could a simpler interpretation might be: coupled models show larger sensitivity to resolution that AMIP models because the SST mean state improves with resolution in the coupled models (whereas it can't in the AMIP ones, because it's prescribed). This suggests the blocking changes with resolution are more to do with mean-state changes than changes in the variability. Do you think the results support this conclusion?

This is plausible and being investigated in a separate study (see comments 3, 30).

35. Figs 1 & 2: perhaps change contour levels so that zero isn't shown, e.g. begin at +/-0.005. It might help reduce the appearance of noisiness in the plots.

We find the degree of noisiness acceptable; all plots contain large contiguous areas of negative blocking biases. The spatially resolved information is further aggregated in over domains in subsequent figures.

36. Figs 3-6: define the different elements of the boxplots (is "box-whisker" plot a more typical name for these?). What is the triangle, quantiles for the box, the circles, etc. Panel labels (a,b,...) could be useful.

Done. We have modified to use the term "box-and-whisker plot" on this occurrence, but then stick to the very common shorthand "boxplot" in the remainder of the manuscript.

37. Captions in general: since so many of the plots are of the same type, it would be useful for just the first plot of each kind to have a full descriptive caption, and then for subsequent plots just say, "As Fig. X, but for [JJA, ANOM, etc]". Otherwise the reader has to go through each caption to find out if much has changed from the previous plot.

Done.

Referenes

Scaife, A. A., Copsey, D., Gordon, C., Harris, C., Hinton, T., Keeley, S., et al. (2011). Improved Atlantic winter blocking in a climate model. *Geophysical Research Letters*, *38*(23), L23703. https://doi.org/10.1029/2011GL049573

Schiemann, R., Demory, M.-E., Shaffrey, L. C., Strachan, J., Vidale, P. L., Mizielinski, M. S., et al. (2017). The Resolution Sensitivity of Northern Hemisphere Blocking in Four 25-km Atmospheric Global Circulation Models. *Journal of Climate*, *30*(1), 337–358. https://doi.org/10.1175/JCLI-D-16-0100.1